# Hypoxia-inducible factor prolyl hydroxylase domain (PHD) inhibition after contusive spinal cord injury does not improve locomotor recovery

George Z. Wei[1,2,3], Sujata Saraswat Ohri[3,4], Nicolas K. Khattar[2,3,4], Adam W. Listerman[3], Catherine H. Doyle[3,4], Kariena R. Andres[3], Saravanan S. Karuppagounder[5,6], Rajiv R. Ratan[5,6], Scott R. Whittemore[2,3,4,7]*, Michal Hetman[1,2,3,4,7]*

1 University of Louisville School of Medicine, Louisville, Kentucky, United States of America, 2 Department of Pharmacology & Toxicology, University of Louisville School of Medicine, Louisville, KY, United States of America, 3 Kentucky Spinal Cord Injury Research Center, University of Louisville School of Medicine, Louisville, KY, United States of America, 4 Department of Neurological Surgery, University of Louisville School of Medicine, Louisville, KY, United States of America, 5 Sperling Center for Hemorrhagic Stroke Recovery, Burke Neurological Institute, White Plains, NY, United States of America, 6 Feil Family Brain and Mind Research Institute, Weill Medical College of Cornell University, New York, NY, United States of America, 7 Department of Anatomical Sciences and Neurobiology, University of Louisville School of Medicine, Louisville, KY, United States of America

* swhittemore@louisville.edu (SRW); michal.hetman@louisville.edu (MH)

**Data Availability Statement:** All relevant analyzed data are within the paper and its Supporting Information files.

## Abstract

Traumatic spinal cord injury (SCI) is a devastating neurological condition that involves both primary and secondary tissue loss. Various cytotoxic events including hypoxia, hemorrhage and blood lysis, bioenergetic failure, oxidative stress, endoplasmic reticulum (ER) stress, and neuroinflammation contribute to secondary injury. The HIF prolyl hydroxylase domain (PHD/EGLN) family of proteins are iron-dependent, oxygen-sensing enzymes that regulate the stability of hypoxia inducible factor-1α (HIF-1α) and also mediate oxidative stress caused by free iron liberated from the lysis of blood. PHD inhibition improves outcome after experimental intracerebral hemorrhage (ICH) by reducing activating transcription factor 4 (ATF4)-driven neuronal death. As the ATF4-CHOP (CCAAT-enhancer-binding protein homologous protein) pathway plays a role in the pathogenesis of contusive SCI, we examined the effects of PHD inhibition in a mouse model of moderate T9 contusive SCI in which white matter damage is the primary driver of locomotor dysfunction. Pharmacological inhibition of PHDs using adaptaquin (AQ) moderately lowers acute induction of *Atf4* and *Chop* mRNAs and prevents the acute decline of oligodendrocyte (OL) lineage mRNAs, but does not improve long-term recovery of hindlimb locomotion or increase chronic white matter sparing. Conditional genetic ablation of all three PHD isoenzymes in OLs did not affect *Atf4*, *Chop* or OL mRNAs expression levels, locomotor recovery, and white matter sparing after SCI. Hence, PHDs may not be suitable targets to improve outcomes in traumatic CNS pathologies that involve acute white matter injury.

**Funding:** This work was supported by The NIH: NS108529 (NINDS), GM10350 (NIGMS), Norton Healthcare, the Commonwealth of Kentucky Research Challenge for Excellence Trust Fund (SRW, MH) and the Leona M. and Harry B. Helmsley Charitable Trust (SRW, SSO, MH) and the Integrated Programs in Biomedical Sciences (GZW).

**Competing interests:** The authors have declared that no competing interests exist.

## Introduction

The pathophysiology of spinal cord injury (SCI) consists of a primary injury phase which occurs as the direct result of mechanical insult to the spinal cord and a secondary injury phase that involves multiple pathophysiological mechanisms including inflammation, vascular disruption, blood hemorrhage and ischemia. These mechanisms further exacerbate the initial injury and lead to greater functional loss [1, 2]. The proteostasis network includes all proteins with a role in protein synthesis, folding, disaggregation, or degradation [3]. Integral to proteostasis are the heat shock response (HSR), the endoplasmic reticulum stress response (ERSR), the integrated stress response (ISR), and the unfolded protein response (UPR) pathways which further attempt to restore cellular homeostasis and if unsuccessful initiate apoptosis and cell death. We and others have shown that the ERSR is activated acutely after SCI [4–9]. The ERSR initially promotes cell survival by reducing global protein synthesis and upregulating chaperones that assist in protein folding. However, excessive or sustained ER stress that cannot be resolved initiates cellular death and contributes to the secondary injury cascade [4, 5, 10, 11]. Interventions that target or alleviate the ERSR after SCI reduce oligodendrocyte (OL) death, protect white matter, and improve locomotor recovery [4, 6–8, 12].

The ERSR drives expression of activating transcription factor 4 (ATF4) and its target and partner CCAAT-enhancer-binding protein homologous protein (CHOP/DDIT3). Upon activation of the ERSR, these transcription factors (TFs) drive a gene expression program that supports amino acid and protein synthesis in an attempt to restore ER homeostasis. However, if ER stress is unresolved, ATF4/CHOP-mediated gene expression becomes cytotoxic by causing ER protein overload, ER-induced oxidative stress, mitochondrial damage, and ultimately apoptosis [13].

The HIF prolyl hydroxylase domain proteins PHD1/EGLN1, PHD2/EGLN2, and PHD3/EGLN3, encoded by the *Egln1*, *Egln2*, and *Egln3* genes, are a class of metalloenzymes oxygen sensors that, under normoxic conditions, hydroxylate proline residues of hypoxia-inducible factor 1α (HIF-1α) to stimulate its proteasomal degradation [14]. Under hypoxic conditions, PHDs are inactive so that HIF-1α accumulates and activates adaptive genes that protect against hypoxia. However, pharmacological PHD inhibition using adaptaquin (AQ) or genetic deletion of the *Egln1-3* genes protect neurons from oxidative death due to blood lysis-derived free iron and improves functional outcome after experimental intracerebral hemorrhage (ICH) [15]. Those beneficial effects appear to be mediated by attenuation of the ATF4-driven cytotoxic gene expression program that is activated in neurons under oxidative stress [15]. PHD-mediated hydroxylation of several ATF4 proline residues is a likely mechanism that promotes cytotoxic activity of ATF4 [15].

Recent studies further demonstrate the protective role of pharmacological PHD inhibition by reducing ATF4-CHOP-mediated neuronal death and protecting against oxidative damage in models of neurodegeneration including Parkinson's and Alzheimer's disease [16–20], where neuronal loss is paramount. However, PHD inhibitors have not been evaluated for their therapeutic potential in protecting white matter against acute injuries such as thoracic contusive SCI where the ATF4 target gene *Chop* plays a major pathogenic role [4]. Furthermore, the role of PHDs in ATF4-CHOP activation after SCI is unknown. The current study was undertaken to determine the contributions of PHDs to ATF4-CHOP activation and white matter loss after SCI.

## Materials and methods

### Animals

All animal procedures were approved by the University of Louisville Institutional Animal Care and Use Committee and the Institutional Biosafety Committee, in accordance with guidelines

from the Public Health Service Policy on Humane Care and Use of Laboratory Animals, Guide for the Care and Use of Laboratory Animals (Institute of Laboratory Animal Resources, National Research Council, 1996), and strictly adhered to NIH guidelines on use of experimental animals. AQ experiments were performed on wild type (WT) 8–10 week-old C57BL/6 female mice (Envigo, Indianapolis, IN) fed a standard *ad libitum* chow and housed under 12h dark/light cycle. Females are predominantly used in rodent SCI literature due to lower incidence of post-operative complications and, therefore, have better survival as compared to males [21]. Moreover, while sex-specific drug effects are possible, locomotor recovery in SCI rodents is not significantly affected by sex [22–24]. Therefore, AQ studies were performed in females to reduce the animal number needed for adequately powered data. *Plp-cre^{ERT2}* (proteolipid protein) (Plp-cre-B6.Cg-Tg (Plp1-Cre/ERT) 3Pop/J; catalog #5975) and *Egln1/2/3^{fl/fl}* (*Egln^{2tm2Fong} Egln^{1tm2Fong} Egln^{3tm2Fong}/J*; Stock No: 028097) mice were acquired from The Jackson Laboratory (Bar Harbor, ME). Those lines were crossed to generate experimental subjects (*Egln1/2/3^{fl/fl}:Plp-Cre^{ERT2}*), which, due to limited availability of animals with the desired genotype, included both males and females. Those mice were treated with tamoxifen to induce *Egln1/2/3* knockout primarily in CNS myelinating OLs [25–27] (see the *Drug treatments* paragraph for more details). Controls included male and female *Egln1/2/3^{fl/fl}:Plp-Cre^{ERT2}* that were treated with vehicle and WT mice that were treated with tamoxifen. Importantly, PLP is expressed earlier than myelin basic protein (MBP) during development, beginning during embryonic life and at early developmental stages of OPC differentiation [25, 28], so in tamoxifen-induced *Egln1/2/3^{fl/fl}:Plp-Cre^{ERT2}* mice, some OPC recombination may be expected as well. Sex unbalanced groups emerged in studies using *Egln1/2/3^{fl/fl}:Plp-Cre^{ERT2}* mice due to limited availability of males of comparable age and peri- and/or post-operative loss of animals (see S1 Table for more details).

## SCI

Prior to surgery, mice were anesthetized using an intraperitoneal injection of 0.4 mg/g body weight Avertin (2,2,2-tribromoethanol in 0.02 ml of 1.25% 2-methyl-2-butanol in saline, Sigma-Aldrich, St. Louis, MO). The back of the mice was shaved and disinfected using a 4% chlorohexidine solution. Lacri-Lube ophthalmic ointment (Allergen, Madison, NJ) was used to prevent drying of the eyes. A dorsal laminectomy was done at the T9 vertebrae and positioned under the Infinite Horizons (IH) Impactor as previously described [4, 29]. A moderate contusion injury (50 kdyn force/400–600 μm displacement) was delivered and mice were immediately placed onto a temperature-controlled 37°C heating pad until sternal time. Mice were administered buprenorphine twice daily for the following 2 days. Gentamycin (50 mg/kg; Boehringer Ingelheim, Ridgefield, CT) was administered subcutaneously to reduce infection. Controls included sham animals that received T9 laminectomy only. All surgeries were performed without knowledge of a group assignment or genotype. For each study, surgeries for all groups were performed on the same day or two consecutive days with random sequence of animals from various groups/genotypes.

## Drug treatments

Tamoxifen was dissolved in corn oil (Sigma, C8267) at 20 mg/mL and administered intraperitoneally (1 mg/day) beginning 21 days prior to SCI and continuing for 8 consecutive days as previously described [30]. Adaptaquin (AQ), obtained from Dr. Rajiv Ratan and Dr. Saravanan Karuppagounder (Burke Neurological Institute, Weill Medical College of Cornell University), was dissolved in a solution of 0.03% DMSO and olive oil. Fresh aliquots were prepared daily for treatment. AQ was administered by intraperitoneal injections (0.1 cc/injection 30

mg/kg) first, within 1 hour after SCI, and then, daily for either 3 days (the experiment to collect tissues at 3 days post injury; the last AQ injection administered 2h before euthanasia) or 7 days (the experiment to determine effects of AQ on locomotor recovery) of treatment. The AQ dosing was based on the previous ICH study which confirmed blood-brain barrier penetration and anti-ATF4 activity in the brain [15]. Proton nuclear magnetic resonance spectroscopy ($^1$H NMR) was used to confirm the molecular integrity of AQ (S1 Fig).

## Behavioral assessment

Animals were habituated to human interaction and handling twice a day for 5 consecutive days, 1 week prior to SCI/Sham. Baseline Basso Mouse Scale (BMS) locomotor scores were obtained prior to injury for every individual animal and weekly following SCI for 6 weeks [4, 31]. Raters were trained by Dr. Basso and colleagues at the Ohio State University and were blinded to the animal genotype and treatment groups. The order of animal analysis was random.

## RNA extraction and analysis

Total RNA was extracted from spinal cord tissue at the injury epicenter (5 mm segment spanning the injury site) using Trizol (Invitrogen) according to the manufacturer's guidelines. RNA was quantified by ultraviolet spectroscopy (NanoDrop2000, Thermo Scientific, Waltham, MA). cDNA synthesis was performed with random hexamers using 500 ng of total RNA using the Invitrogen SuperScript IV VILO Master Mix I (Thermo Fisher) in a 20 μL reaction volume. All cDNAs were diluted 10x with water before using as a template for quantitative real time RT-PCR (qPCR). qPCR was performed using ViiA 7 system (Applied Biosystems, Foster City, CA). Briefly, diluted cDNAs were added to TaqMan universal PCR master mix or SYBR Green master mix (Applied Biosystems) and run in triplicate. Primer sets are listed in S2 Table. For qPCR analysis of PHD family mRNAs, protocols and primers were used as previously described [32, 33]. RNA levels were quantified using the ΔΔCT method with *Gapdh* as a reference gene. Transcript levels were normalized to their respective levels in sham or vehicle controls and expressed as fold-changes.

## White matter sparing (WMS)

WMS was evaluated as described previously [4, 34]. Six weeks post-SCI, mice were anesthetized with avertin, followed by thoracotomy and transcardial perfusion with ice-cold PBS and 4% paraformaldehyde (PFA). Spinal cords were dissected and submerged in 4% PFA overnight at 4˚C. They were then transferred to 30% sucrose for at least 7 days at 4˚C, blocked in Tissue Freezing Media (Cat # 72592, Electron Microscopy Science, Hatfield, PA) and stored at -20˚C. Spinal cords were serially cut (20 μm) in transverse sections spanning 5 mm rostral and caudal to the injury epicenter, and were stained for myelin using iron eriochrome cyanine (EC) with an alkali differentiator [35]. Images were captured using a Nikon Eclipse Ti inverted microscope (Nikon Instruments Inc., Melville NY), and white matter was traced using Nikon Elements software. The epicenter of each injury was identified visually based on the section with the least amount of spared white matter. Data were normalized to spared white matter in corresponding non-injured sections. All imaging and analyses were performed blinded to ensure unbiased quantification.

## ODD-luciferase assay

The biological activity of AQ used in animal studies was confirmed on cultured SH-SY5Y cells using the HIF-1α oxygen degradation domain (ODD)-luciferase reporter assay (S2 Fig) as previously described [15, 36].

## Statistical analysis

All qPCR or image analysis data (WMS) were analyzed using the non-parametric Mann-Whitney test (*u*-test, single sided). Repeated-measures ANOVA (RM-ANOVA) followed by Tukey post hoc tests were used for analyzing BMS locomotor recovery data. *A priori* power calculations were performed for analysis of locomotor recovery. Power analysis based on BBB/BMS variance in published rodent SCI studies with standard deviations between 1.5–2.3 shows that the ability to detect a significant difference of 10% in BMS with at least 90% power in a sample size of 8/group. Hence, all locomotor recovery assessments were adequately powered. Data are reported as mean ± SD. Statistical analyses were performed using SPSS, version 25 (IBM).

## Results

At 3 days post-SCI, there were similar declines in neuron-specific enolase (*Nse/Eno2*) mRNA and increases in astrocyte-specific mRNAs (*Gfap*, *Glul*) in vehicle- and AQ-treated animals suggesting no effects of AQ on either cell type (Fig 1A). However, there were increased OL-specific transcripts (*Olig2*, *Mbp*), along with decreased mRNAs of *Atf4*, *Chop*, and the ATF4/CHOP target gene *Trib3* in AQ-treated animals (Fig 1B). Moreover, *Map2* mRNA, which is expressed mainly in neurons, but also detected in OLs [37], was increased. These results suggest reduced acute loss of OLs and attenuated activation of the ATF4-CHOP pathway. However, as expression of the ER stress-associated ATF4/CHOP targets *Gadd34/Ppp1r15a* and *Slc7a11* was unaffected, the anti-ATF4 effects of AQ appear to be target gene-specific. Finally, AQ treatment did not modify ERSR-associated induction of *Grp78* and *Xbp1* mRNAs suggesting no general attenuation of the ERSR.

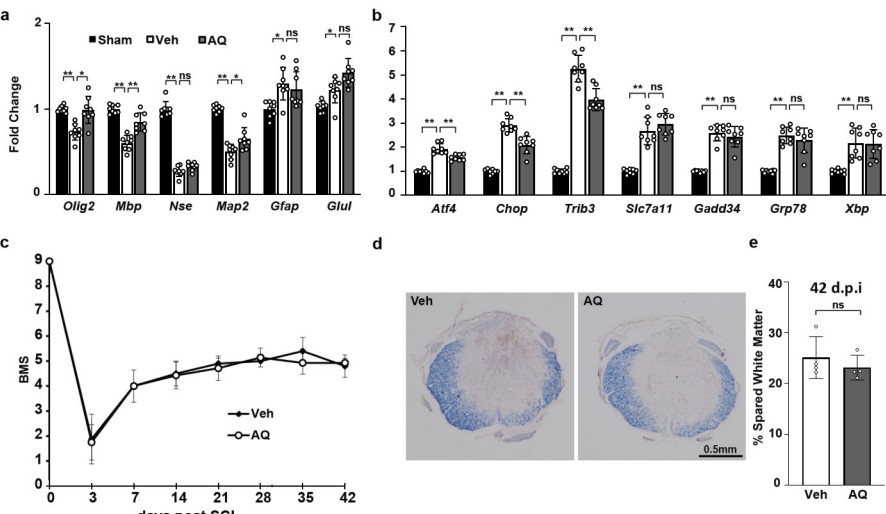

**Fig 1. After SCI, AQ attenuates the acute loss of OL lineage mRNAs, moderately reduces acute ATF4 and CHOP activation, but does not improve chronic functional locomotor recovery.** (a) neural cell-specific and (b) ISR/ERSR and ATF4-regulated gene mRNA levels 72 hours-post SCI. Transcript levels (normalized to *Gapdh*) are expressed as fold change sham controls. Data in (a, b) are the mean ± SD (n = 8, *p<0.05; **p<0.01; ns, p>0.05, *u*-test). (c) BMS analysis of hindlimb locomotion revealed no significant differences in hindlimb locomotor recovery between vehicle (Veh)- (n = 6) and AQ-treated 30 mg/kg (n = 7) mice (repeated measure two-way ANOVA showed significant effects of time after injury /$F_{5,21}$ = 12.34; p<0.001/ but no significant effects of treatment /$F_{1,59}$ = 0.03; p>0.05/ or the interaction between time after injury and treatment /$F_{5,21}$ = 0.97, p>0.05/; see S3 Table for raw BMS data). (d) Representative images of the injury epicenter stained with EC to identify myelin (6 weeks post-SCI) (e) Quantitative analysis of EC-stained sections through the injury epicenter shows similar extent of spared white matter. Data are the mean ± SD (n = 4, p>0.05, *u*-test).

To determine the effects of AQ treatment on chronic locomotor recovery, mice received vehicle or 30 mg/kg AQ immediately after injury and then daily for 7 days. Hindlimb function was evaluated using the BMS for 6 weeks post-SCI. Comparison of BMS scores between vehicle- and AQ-treated mice revealed no significant differences (Fig 1C). Consistent with no AQ effects on locomotor recovery, a similar extent of white matter sparing was detected in AQ- and vehicle-treated groups (Fig 1D and 1E). These data show that despite transient improvement in OL survival acutely after contusive SCI, pharmacological inhibition of PHDs using AQ did not increase chronic white matter sparing or locomotor recovery.

To determine if the genetic deletion of the PHDs in OLs affects outcome after SCI, we used OL-specific *Plp-cre^{ERT2+/+}:Egln1/2/3^{fl/fl}* mice to conditionally remove all 3 PHD isoenzymes in OLs following treatment with tamoxifen. Fourteen days after the completion of tamoxifen or vehicle treatment, we confirmed that tamoxifen treatment resulted in lower *Phd/Egln* mRNA levels in uninjured spinal cords of *Plp-cre^{ERT2+/+}:Egln1/2/3^{fl/fl}* mice (Fig 2A). Consistent downregulation of *Phd/Egln* transcripts ranged from 23 to 25% of control values. These findings suggest that (*i*) gene deletion efficiency was similar for each isoform and (*ii*) the deletion occurred in most PLP-positive, mature OLs as their estimated spinal cord content is 20–25% [38–40] and at least the maximally expressed *Phd2/Egln2* appears to be ubiquitously present in all spinal cord cells [41]. In addition, *Phd–Egln* mRNAs were downregulated by 25–32% in the OL-rich tissue of the optic nerve (Fig 2A), but not the OL-lacking liver of tamoxifen-treated *Plp-cre^{ERT2+/+}:Egln1/2/3^{fl/fl}* mice (S3 Fig). The OL-specific loss of *Phd/Egln* genes was biologically relevant as the established HIF target genes *Vegf* and *Epo* were upregulated, including 27% increase of *Vegf* or 37–39% increases of *Vegf* and *Epo* transcripts in the spinal cord or the optic nerve, respectively. Such upregulation is expected as a result of reduced negative regulation of HIF due to PHD deficiency [15, 42]. The lesser response in the spinal cord than the optic nerve is likely caused by high basal level spinal neuron expression of *Epo* and *Vegf* mRNAs [41] which can potentially dilute the OL induction of those genes after *Phd/Egln* deletion. Thus, our data suggest OL-specific knockout of PHD1/2/3 isoforms.

At 72 h after SCI, tamoxifen- and vehicle-treated *Plp-cre^{ERT2+/+}:Egln1/2/3^{fl/fl}* mice had similar neural cell mRNA levels, suggesting unaffected acute loss of neurons and OLs (Fig 2B). No effects on the expression of *Atf4*, *Chop*, *Gadd34* or *Grp78* mRNAs were observed (Fig 2C). Moreover, comparison of 6-week BMS scores for tamoxifen- and vehicle-treated *Plp-cre^{ERT2+/+}:Egln1/2/3^{fl/fl}* mice revealed no significant differences in locomotor recovery (Fig 2D). Likewise, no differences in chronic hindlimb locomotor recovery were observed when tamoxifen-induced OL-PHD knockouts were compared to WT mice that received identical tamoxifen treatment (Fig 2D). Therefore, OL-selective deletion of HIF-PHD does not affect SCI-associated acute OL loss, ATF4-CHOP signaling or chronic locomotor recovery.

## Discussion

Adaptaquin (AQ), a hydroxyquinoline-based inhibitor of PHDs, abrogates ATF4-CHOP-dependent neuronal death and improves functional outcomes in mouse models of ICH and Parkinson's disease [15, 19]. While these studies implicate PHDs in ATF4-CHOP-mediated neuronal death, their involvement in OL death and white matter damage after SCI remains unknown. Acutely after SCI, the pro-apoptotic, ERSR-activated transcription factors ATF4 and CHOP are upregulated in neurons and OLs [4–8]. Our previous study showed improved behavioral outcome after thoracic contusive SCI in *Chop^{-/-}* mice that was associated with increased WMS and attenuated acute loss of OLs [4]. Likewise, pharmacological interventions targeting proteostasis attenuated SCI-associated activation of ATF4-CHOP while improving functional recovery and WMS [7]. Therefore, we hypothesized that pharmacological and/or

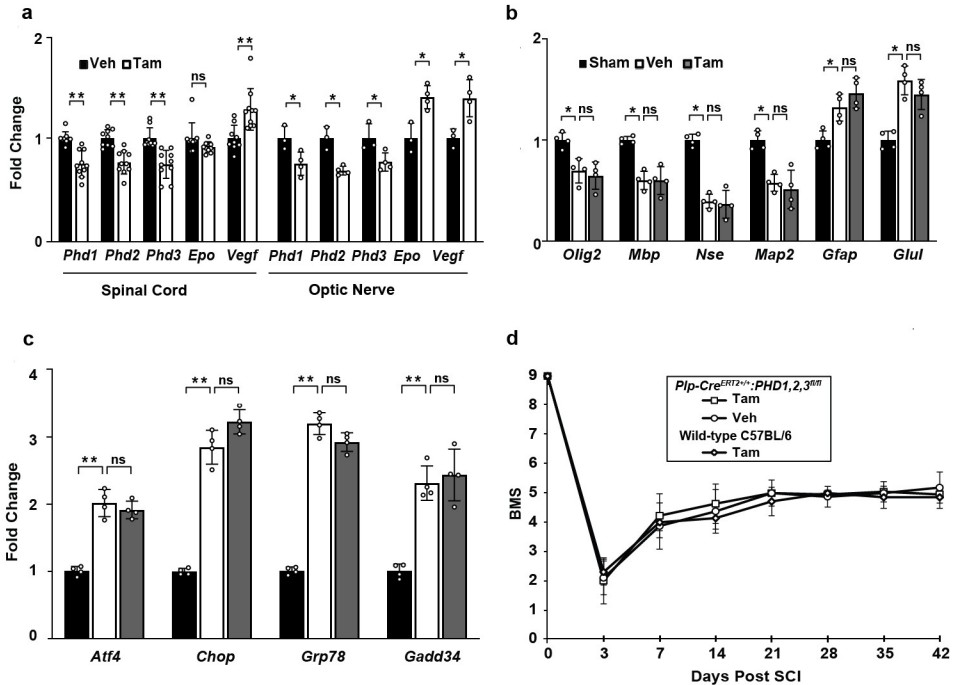

**Fig 2. Effects of HIF-PHDs deletion in OL lineage cells after SCI.** (a) Tamoxifen (Tam)-treated *Plp-cre*<sup>ERT2+/+</sup>:*Egln1/2/3*<sup>fl/fl</sup> mice (n = 10, M:F = 4:6) show reduced *Phd1/Egln1*, *Phd2/Egln2*, and *Phd3/Egln3* mRNAs in the spinal cord (controls received vehicle /Veh/, n = 9, M:F = 4:5). Similarly, Tamoxifen (Tam)-treated *Plp-cre*<sup>ERT2+/+</sup>:*Egln1/2/3*<sup>fl/fl</sup> mice (n = 4, M:F = 2:2) show reduced *Phd1/Egln1*, *Phd2/Egln2*, and *Phd3/Egln3* in the optic nerve (controls received vehicle /Veh/, n = 3, M:F = 2:1). Confirming biological relevance of the OL-specific PHD1/2/3 deficiency, established HIF target genes including *Vegf* or *Vegf* and *Epo* were upregulated in OL-containing spinal cord and optic nerve, respectively. (b) At 72 h after injury, SCI-associated declines of neuronal- or OL mRNAs are unaffected by Tam-mediated OL-selective deletion of *PHD1/2/3*. (c) Likewise, Tam-mediated knockout of OL-*PHD1/2/3* did not attenuate SCI-associated induction of ERSR transcripts including *Atf4* and *Chop*. Transcript levels (normalized to *Gapdh*) are expressed as fold changes of Veh control (a) or sham *Plp-cre*<sup>ERT2+/+</sup>:*Egln1/2/3*<sup>fl/fl</sup> controls (b,c). Data (b,c) are the mean ± SD (n = 4, M:F = 2:2 *p<0.05; ns, p>0.05, *u*-test). (d) BMS analysis of hindlimb locomotion revealed no significant differences in recovery between Veh- (n = 8, M:F = 2:6), Tam-treated *Plp-cre*<sup>ERT2+/+</sup>:*Egln1/2/3*<sup>fl/fl</sup> mice (n = 12, M:F = 5:7), and Tam-treated WT C57BL/6 mice (n = 10, M:F = 4:6). When comparing Veh- and Tam-treated *Plp-cre*<sup>ERT2+/+</sup>:*Egln1/2/3*<sup>fl/fl</sup> mice, repeated measure two-way ANOVA showed significant effects of time after injury ($F_{5,56}$ = 11.89, p<0.001), but no significant effects of Tam treatment ($F_{2,95}$ = 2.6, p>0.05), or the interaction between the time after injury and Tam treatment ($F_{10,56}$ = 0.99, p>0.05); see S3 Table for raw BMS data.

genetic suppression of ATF4-CHOP signaling after thoracic contusive SCI would similarly restore proteostasis and prevent chronic OL loss, leading to increased spared white matter and improved functional outcomes. However, AQ-mediated attenuation of acute OL loss and ATF4-CHOP signaling did not improve chronic locomotor recovery or WMS. Importantly, although AQ reduced *Atf4*, *Chop*, and *Trib3* mRNA levels, there was no effect on three other ERSR transcripts, *Gadd34*, *Grp78*, and *Xbp1* (Fig 1B). In previous studies, where increased acute OL/OPC mRNA levels were accompanied by increases in chronic locomotor recovery, there was reduced activation of not only *Atf4* and *Chop*, but also other ERSR transcripts [4, 7, 12, 30, 43]. It is likely that AQ inhibits ATF4-CHOP signaling that is distinct from their involvement in the ERSR and therefore limits the effects of AQ's effects on SCI outcome. Indeed, in the ICH model, AQ was proposed to inhibit the ATF4-mediated iron-dependent ferroptosis, but not the cytotoxic ER stress [15, 44]. Hence, our AQ studies suggest that although PHDs contribute to ATF4-CHOP signaling in the contused spinal cord, the cytotoxic ERSR is the dominant signaling pathway that drives chronic white matter damage and

functional deficits [4, 9]. This interpretation is further supported by results from SCI mice with OL/OPC-specific PHD deletion (Fig 2). OLs are uniquely sensitive to injury-induced ERSR because of their high protein translation requirements [4, 45–48]. However, no effects on thoracic contusive SCI-associated increases of ATF4 and CHOP signaling were observed in the OL-specific *Phd/Egln*[1,2,3-/-] mice. Likewise, acute OL loss and locomotor recovery were unaffected.

These observations are in contrast to those from various CNS injury models with primary effects on the grey matter [16–20]. Those data suggest that PHD inhibition might improve functional outcomes in lumbar or cervical models of SCI, where neuronal death is paramount to loss of function [49–51]. The differences in the effects of AQ on ICH and thoracic contusive SCI highlight the complexities of the proteostasis network [3]. Following CNS trauma, multiple proteostasis signaling pathways are differentially activated by distinct pathophysiological stimuli. Depending on the extent and duration of those stimuli, the various pro-homeostatic and pro-apoptotic signals differentially summate. Moreover, there are both overlapping and diverse aspects of the respective arms of proteostasis signaling [52–55]. Unfortunately, there is currently no consensus on how best to globally target proteostasis effectors for treating CNS trauma. While it is likely to be therapeutically beneficial, it will have to be empirically determined for each type of injury.

## Supporting information

**S1 Fig. [1]H NMR spectrum for the AQ batch used in the current SCI study.** [1]H NMR (DMSO-$d_6$, 500 MHz) δ 9.84 (s, 1H), 8.84 (dd, 1H), 8.2–8.4 (m, 1H), 7.66 (d, 1H), 7.54 (dd, 1H), 7.3–7.5 (m, 6H), 6.8–6.9 (m, 2H), 6.4–6.5 (m, 2H) matches the spectra of AQ as previously described [56]. Data were recorded on a Varian Unity Inova 500 MHz and chemical shifts are reported in ppm using the solvent as an internal standard (DMSO-$d_6$ at 2.5 ppm). (TIF)

**S2 Fig. Validation of biological activity for the AQ batch (AQ /KY/) used in the current SCI study.** SH-SY5Y ODD-Luc cells were treated with increasing concentrations of AQ (0.1 μM-10 μM) for 3 h. ODD-luciferase activity was determined by luminometry. AQ inhibits HIF prolyl hydroxylase activity and stabilize luciferase fused to the oxygen degradation domain (ODD). (TIF)

**S3 Fig. Effects of OL-specific HIF-PHDs deletion in the OL-lacking liver tissue.** Tamoxifen (Tam)-treated *Plp-cre*[ERT2+/+]:*Egln1/2/3*[fl/fl] mice (n = 4, M:F = 2:2) show no changes in *Phd1/Egln1*, *Phd2/Egln2*, *Phd3/Egln3*, *Epo*, and *Vegf* mRNAs (controls received vehicle /Veh/, n = 3, M:F = 2:1). (TIF)

**S1 Table. Experimental design.** (DOCX)

**S2 Table. List of qPCR primers.** (DOCX)

**S3 Table. Raw BMS score data for individual mice.** (DOCX)

## Acknowledgments

The authors wish to thank Dr. Lukasz Slomnicki for discussions, Christine Yarberry for performing surgical procedures, Darlene A. Burke for help with statistical analyses, Johnny Morehouse and Jason Beare for BMS analyses and Molly Parsch for her excellent technical assistance.

## Author Contributions

**Conceptualization:** George Z. Wei, Sujata Saraswat Ohri, Rajiv R. Ratan, Scott R. Whittemore, Michal Hetman.

**Formal analysis:** George Z. Wei, Saravanan S. Karuppagounder, Michal Hetman.

**Funding acquisition:** Sujata Saraswat Ohri, Scott R. Whittemore, Michal Hetman.

**Investigation:** George Z. Wei, Sujata Saraswat Ohri, Nicolas K. Khattar, Adam W. Listerman, Catherine H. Doyle, Kariena R. Andres, Saravanan S. Karuppagounder.

**Resources:** Saravanan S. Karuppagounder, Rajiv R. Ratan.

**Supervision:** Rajiv R. Ratan, Scott R. Whittemore, Michal Hetman.

**Visualization:** George Z. Wei.

**Writing – original draft:** George Z. Wei.

**Writing – review & editing:** Sujata Saraswat Ohri, Saravanan S. Karuppagounder, Rajiv R. Ratan, Scott R. Whittemore, Michal Hetman.

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
