## [Decision Letter · Decision Letter 0]

7 Oct 2020

PONE-D-20-28946

Hypoxia-inducible factor prolyl hydroxylase domain (PHD) inhibition after contusive spinal cord injury does not improve locomotor recovery

PLOS ONE

Dear Dr. Hetman,

Thank you for submitting your manuscript to PLOS ONE. After careful consideration, we feel that it has merit but does not fully meet PLOS ONE’s publication criteria as it currently stands. Therefore, we invite you to submit a revised version of the manuscript that addresses the points raised during the review process.

The critiques below indicate the required changes.  The authors are encouraged to address the issue of knockdown of PDH, as reviewer 2 notes.

We look forward to receiving your revised manuscript.

Kind regards,

Kimberly R. Byrnes, Ph.D.

Academic Editor

PLOS ONE

Journal Requirements:

Reviewers' comments:

Reviewer's Responses to Questions

**Comments to the Author**

1. Is the manuscript technically sound, and do the data support the conclusions?

Reviewer #1: Yes

Reviewer #2: Yes

2. Has the statistical analysis been performed appropriately and rigorously? 

Reviewer #1: Yes

Reviewer #2: Yes

3. Have the authors made all data underlying the findings in their manuscript fully available?

Reviewer #1: Yes

Reviewer #2: Yes

4. Is the manuscript presented in an intelligible fashion and written in standard English?

Reviewer #1: Yes

Reviewer #2: Yes

5. Review Comments to the Author

Reviewer #1: The current studies by Wei et al. were designed to assess effects of prolyl hydroxylase domain (PHD) inhibition in a mouse model of moderate T9 contusive SCI. The authors reported that pharmacological inhibition of PHDs by adaptaquin moderately reduced Atf4 and Chop mRNAs expression levels and increased oligodendrocyte lineage mRNAs. However, the treatment did not affect locomotor function deficit and tissue damage induced by SCI. Furthermore, genetic deletion of all three PHD iso-enzymes in oligodendrocyte lineage did not affect Atf4, Chop or OL mRNA expression levels, or locomotor recovery after SCI. They concluded that PHDs may not be suitable targets to treatment SCI. Overall, this is the first study that demonstrates no effects of PHD inhibition in SCI model. In general, the manuscript is well written and the study is well designed. The authors provide appropriate and rigorous analysis and the conclusions are balanced with relevant literature. Methods are clear and descriptive of the procedures allowing reproduction of the experiments. Statistical analysis is properly applied by the authors. I have only few minor concerns:

1. Please justify using the literature the use of only female mice in the study for AQ experiments (Fig 1). More discussion is needed regarding the narrow focus of the study since males were not included in this study.

2. In the HIF-PHDs deletion study (Fig 2)---What is the rationale for using such different male/female ratio for each of the various tests?

3. For the statistical analysis, please specify if power analysis and exclusion of data were included in this research.

4. The authors should show individual data points where possible.

Reviewer #2: Wei et al. describe the effect of inhibiting the PHD family proteins on the outcomes of contusive SCI in the mouse model. They demonstrate that while pharmacological PHD inhibition moderately decreases levels of ER stress markers it has no effect on either histological or functional outcomes. Similarly, genetic oligodendrocyte-specific inhibition of Phd1-3 does not affect SCI outcomes.

In general, the manuscript is well written, the experiments are appropriately executed, analyzed and interpreted.

The only major issue is the extent of Phd1-3 knock out following tamoxifen treatment. While data in Fig 2a indicate that knock out of each of the genes significantly lowers its mRNA expression, the degree of this decrease is very small (10-15%). Statistically significant does not necessarily equal biologically relevant. Since these genes are expressed in other cell types, the authors should check expression in purified oligodendrocytes. It is also possible that expression would be lower at protein rather than mRNA level.

Minor issues:

• Please specify PHD family members in the introduction including their proper gene names (Egln1-3).

• AQ treatments were given for 3 days. Please provide better justification for timing - longer treatment (7-10 days) is often used for SCI.

6. PLOS authors have the option to publish the peer review history of their article (what does this mean?). If published, this will include your full peer review and any attached files.

Reviewer #1: No

Reviewer #2: No

---

## [Author Response · Author response to Decision Letter 0]

23 Feb 2021

Response to Reviewers 

Re: Submission titled <Hypoxia-inducible factor prolyl hydroxylase domain (PHD) inhibition after contusive spinal cord injury does not improve locomotor recovery>

Rev #1:

1. Please justify using the literature the use of only female mice in the study for AQ experiments (Fig 1). More discussion is needed regarding the narrow focus of the study since males were not included in this study.

Response: Females are predominantly used in rodent SCI literature due to lower incidence of urinary tract infections and better survival as compared to males /PMID: 32849242/. Furthermore, there have been multiple studies that show that there is no significant sex difference in locomotor functional recovery after SCI /for instance see, PMID: 19831737 and 32486893/. Therefore, we have used females in AQ studies to minimize the animal number needed for adequate statistical power. Had there been any differences between AQ- and vehicle treated animals, a follow up study using males would have been added. We expanded the Animal paragraph of the Methods section justifying such an experimental design including citing the supporting literature. 

2. In the HIF-PHDs deletion study (Fig 2)---What is the rationale for using such different male/female ratio for each of the various tests?

Response: To properly power experiments that involved quadruple transgenic Egln1/2/3fl/fl :Plp-CreERT2 mice (Fig. 2), we used both sexes. However, sex unbalanced groups emerged in our studies on Egln1/2/3fl/fl : Plp-CreERT2 mice due to limited availability of males of comparable age and their higher mortality due to complications after SCI (see #1, above). The aforementioned expansion of the Methods sections presents that rationale. We also added a Supplementary Table S2 that lists all the studies and provides detailed information about mortality. 

3. For the statistical analysis, please specify if power analysis and exclusion of data were included in this research.

Response: Design of SCI studies to determine effects on locomotor outcome was based on a priori power analysis. We have revised the Methods section (the Statistical Analysis paragraph) to include the description of the underlying power analysis that guided the experimental design). In longitudinal assessments of locomotor recovery, we excluded all data obtained from animals that died or required euthanasia before completion of all assessments (S2 Table). 

4. The authors should show individual data points where possible.

Response: Individual data points for Fig. 1 and Fig. 2 have been added.

Rev #2:

1. The only major issue is the extent of Phd1-3 knock out following tamoxifen treatment. While data in Fig 2a indicate that knock out of each of the genes significantly lowers its mRNA expression, the degree of this decrease is very small (10-15%). Statistically significant does not necessarily equal biologically relevant. Since these genes are expressed in other cell types, the authors should check expression in purified oligodendrocytes. It is also possible that expression would be lower at protein rather than mRNA level.

Response: We agree that ensuring high penetration of OL-specific PHD/EGLN deletion is essential for interpretation of our findings shown in Fig. 2. While analyzing PHD expression in purified OLs would be optimal, isolating mature OLs from adult mouse spinal cord is technically challenging (we have unsuccessfully tried that approach using the commercial Miltenyi reagents). Sufficient numbers of intact OLs cannot be isolated from adult mouse spinal cord as the dissociation procedures kill the majority of these cells. Another approach could be immunostaining. Unfortunately, we were unable to obtain specific staining with any of the available anti-PHD antibodies. Therefore, for the current revision, we have expanded the qPCR validation of the gene deletions as follows:

- to ensure that Phd/Egln mRNA declines in the spinal cord tissue of tamoxifen-treated mice are consistent, we added additional animals increasing n to10 for the OL-KO group (tamoxifen treated) and to 9 for the WT group (vehicle treated). With those bigger groups, we observed consistent downregulation of spinal cord Phd /Egln transcripts. Moreover, the extent of downregulation was similar between various Phd/Egln isoforms (ranging from 23 to 25% controls). Hence, those findings suggest that (i) gene deletion efficiency was similar for each isoform, (ii) the deletion occurred in most PLP-positive, mature OLs as their estimated spinal cord content is 20-25% and as at least the maximally expressed Phd2/Egln2 appears to be ubiquitously present in all spinal cord cells (Allen Brain Atlas; OL content estimation is based on observations that the total C57Bl6 mouse CNS content of all OL linage cells (including OPCs and mature OLs) does not exceed 30% even in such white matter-rich areas as the brainstem /PMID: 30425626/ and that OPCs make 5-10% of all rodent CNS cells /PMID: 14572468/). Those new data are shown in the revised Fig. 2a. Furthermore, we show no changes in Phd1-3, Epo, and Vegf in the OL-lacking livers of tamoxifen-treated Plp-creERT2+/+:Egln1/2/3fl/fl mice (S3 Fig.).

- to further confirm that such mRNA declines represent OL-specific deletion of Phds/Eglns, we also analyzed Phd transcripts in the optic nerve which represents another CNS structure highly enriched for the white matter. Again, Phd/Egln mRNAs were reduced by 25-32% as expected for highly efficient, simultaneous deletion of all 3 genes in mature OLs. Those data are shown in the revised Fig. 2a. 

-to ensure biological relevance of those deletions we also analyzed expression of Vegf and Epo which are positively regulated by HIF, the established downstream target of PHDs/EGLNs. In the OL-KO spinal cord, Vegf was up by 27%; in the OL-KO optic nerve, both Vegf and Epo were up by 38 and 39%, respectively. Hence, response of HIF target genes to OL-specific deletions of PHDs/EGLNs further supports efficient and biologically consequential KO of OL PHDs. Lesser response in the spinal cord than the optic nerve is expected as spinal cord neuron cell bodies contain most of spinal cord Epo and Vegf mRNAs under baseline conditions (Allen brain atlas). Hence, neuronal Epo/Vegf mRNAs likely dilute the OL induction of those genes in response to OL-specific deletion of Phds/Eglns. 

2. Please specify PHD family members in the introduction including their proper gene names (Egln1-3).

Response: We have defined the PHD family members as PHD1/EGLN1, PHD2/EGLN2, and PHD3/EGLN3 in the introduction as requested. 

3. AQ treatments were given for 3 days. Please provide better justification for timing - longer treatment (7-10 days) is often used for SCI.

Response: We performed 2 distinct studies using AQ. In the first study (to collect spinal cord tissue for mRNA analysis at dpi 3), AQ was administered for 3 days with the last administration 2 h before euthanasia and tissue collection. In the second study (to analyze AQ effects on locomotor recovery), the treatment was continued for 7 days. We revised the Methods sections to make that clear.

---

## [Decision Letter · Decision Letter 1]

22 Mar 2021

Hypoxia-inducible factor prolyl hydroxylase domain (PHD) inhibition after contusive spinal cord injury does not improve locomotor recovery

PONE-D-20-28946R1

Dear Dr. Hetman,

We’re pleased to inform you that your manuscript has been judged scientifically suitable for publication and will be formally accepted for publication once it meets all outstanding technical requirements.

Kind regards,

Kimberly R. Byrnes, Ph.D.

Academic Editor

PLOS ONE

Additional Editor Comments (optional):

Reviewers' comments:

Reviewer's Responses to Questions

**Comments to the Author**

1. If the authors have adequately addressed your comments raised in a previous round of review and you feel that this manuscript is now acceptable for publication, you may indicate that here to bypass the “Comments to the Author” section, enter your conflict of interest statement in the “Confidential to Editor” section, and submit your "Accept" recommendation.

Reviewer #1: All comments have been addressed

Reviewer #2: All comments have been addressed

2. Is the manuscript technically sound, and do the data support the conclusions?

Reviewer #1: Yes

Reviewer #2: Yes

3. Has the statistical analysis been performed appropriately and rigorously? 

Reviewer #1: Yes

Reviewer #2: Yes

4. Have the authors made all data underlying the findings in their manuscript fully available?

Reviewer #1: Yes

Reviewer #2: Yes

5. Is the manuscript presented in an intelligible fashion and written in standard English?

Reviewer #1: Yes

Reviewer #2: Yes

6. Review Comments to the Author

Reviewer #1: I believe that all my concerns from my prior review have been adequately addressed and that all responses meet formatting specifications.

Reviewer #2: (No Response)

7. PLOS authors have the option to publish the peer review history of their article (what does this mean?). If published, this will include your full peer review and any attached files.

Reviewer #1: No

Reviewer #2: No

---

## [Editor Report · Acceptance letter]

25 Mar 2021

PONE-D-20-28946R1 

Hypoxia-inducible factor prolyl hydroxylase domain (PHD) inhibition after contusive spinal cord injury does not improve locomotor recovery 

Dear Dr. Hetman:

I'm pleased to inform you that your manuscript has been deemed suitable for publication in PLOS ONE. Congratulations! Your manuscript is now with our production department. 

Kind regards, 

on behalf of

Dr. Kimberly R. Byrnes 

Academic Editor

PLOS ONE